# Quality of Life, Clinical, and Patient-Reported Outcomes after Pencil Beam Scanning Proton Therapy Delivered for Intracranial Grade WHO 1–2 Meningioma in Children and Adolescents

**DOI:** 10.3390/cancers15184447

**Published:** 2023-09-06

**Authors:** Marta García-Marqueta, Miriam Vázquez, Reinhardt Krcek, Ulrike L. Kliebsch, Katja Baust, Dominic Leiser, Michelle van Heerden, Alessia Pica, Gabriele Calaminus, Damien C. Weber

**Affiliations:** 1Center for Proton Therapy, Paul Scherrer Institute, ETH Domain, 5232 Villigen, Switzerland; marta.garcia@psi.ch (M.G.-M.);; 2Department of Radiation Oncology, Inselspital, Bern University Hospital, University of Bern, 3012 Bern, Switzerland; 3Department of Pediatric Hematology and Oncology, University Hospital Bonn, 53127 Bonn, Germany; 4Department of Radiation Oncology, University of Zürich, 8091 Zürich, Switzerland

**Keywords:** intracranial meningioma, children, adolescents, teenagers, proton therapy, pencil beam scanning, quality of life, patient-reported outcomes

## Abstract

**Simple Summary:**

Meningioma is a rare entity in the pediatric and adolescent population. While maximal resection remains the established standard of care, the role of radiation therapy remains unclear. The aim of this retrospective study was to report the clinical and patient-reported outcomes of a cohort of 10 children and adolescents with intracranial meningioma treated with Pencil Beam Scanning Proton Therapy (PBS-PT) at the Paul Scherrer Institute between 1996 and 2022. The local control rates at three and five years were modest, yet overall survival remained excellent. Some patients reported functional status limitation during first year and 2 years after PBS-PT, with only one reporting limitation after 3 years. The good tolerance of the treatment in terms of acute toxicity and the absence of severe long-term side effects support the safety of the treatment and viability of PBS-PT as a suitable therapeutic option for intracranial meningioma in the pediatric and adolescent population.

**Abstract:**

Purpose: The purpose of this study was to report the clinical and patient-reported outcomes of children and adolescents with intracranial meningioma treated with pencil beam scanning proton therapy (PBS-PT). Material and methods: Out of a total cohort of 207 intracranial meningioma patients treated with PBS-PT between 1999 and 2022, 10 (4.8%) were children or adolescents aged < 18 years. Median age was 13.9 years (range, 3.2–17.2). Six (60%) children were treated as primary treatment (postoperative PT, n = 4; exclusive PT, n = 2) and four (40%) at the time of tumor recurrence. Acute and late toxicities were registered according to Common Terminology Criteria of Adverse Events (CTCAE). Quality of life (QoL) before PBS-PT was assessed using PEDQOL questionnaires. Educational, functional, and social aspects after PT were assessed through our in-house developed follow-up surveys. Median follow-up time was 71.1 months (range, 2.5–249.7), and median time to last questionnaire available was 37.6 months (range, 5.75–112.6). Results: Five (50%) children developed local failure (LF) at a median time of 32.4 months (range, 17.7–55.4) after PBS-PT and four (80%) were considered in-field. One patient died of T-cell lymphoma 127.1 months after PBS-PT. Estimated 5-year local control (LC) and overall survival (OS) rates were 19.4% and 100.0%, respectively. Except for one patient who developed a cataract requiring surgery, no grade ≥3 late toxicities were reported. Before PT, patients rated their QoL lower than their parents in most domains. During the first year after PT, one child required educational support, one needed to attend to a special school, one had social problems and another three children required assistance for daily basic activities (DBA). Three years after PT, only one child required assistance for DBA. Conclusions: The outcome of children with intracranial meningioma treated with PBS-PT is in line with other centers who have reported results of radiation therapy delivered to this particular patient group. This therapy provides acceptable functional status profiles with no high-grade adverse radiation-induced events.

## 1. Introduction

Meningioma is the most common non-malignant brain tumor, representing up to 37% of all primary central nervous system (CNS) tumors [1]. Nonetheless, it constitutes a rare and unique entity in the pediatric and adolescent age groups, accounting for 2% of all CNS tumors in children between 0–14 years and 5% between the ages of 15 and 19 years [2,3].

Although the management strategies are often extrapolated from studies conducted in the adult population, several publications have highlighted substantial differences in childhood meningiomas compared to the adult setting in terms of epidemiology, clinical behavior, pathology, and molecular features [4,5,6,7]. Furthermore, most of the available literature on the topic analyzes the prognosis following resection [8,9,10,11,12,13,14], which nowadays remains the standard treatment where achieving gross total resection is the main goal.

Due to the low incidence of pediatric meningioma, there is a notable lack and heterogeneity in the available data concerning the role of radiation therapy (RT). Since children are vulnerable to the potential adverse effects of radiation, the trend in past decades has been to avoid or postpone RT [15,16,17]. Notwithstanding, it has been suggested by some authors that radiation therapy may play a vital role in unresectable or incompletely resected malignant meningioma or as a salvage treatment for recurrent or progressive disease [7,18,19]. The existing pediatric guidelines by the Children’s Cancer and Leukaemia Group (CCLG) recommend evaluating the indication for RT based on a multidisciplinary assessment. Factors such as age, number of recurrences, the feasibility of further surgical procedures, and the risk of recurrence associated with tumor grade and/or aggressive behavior should be taken into consideration [20,21].

Stereotactic radiosurgery (SRS) such as Gamma Knife (GK) or CyberKnife, has been favored as a treatment option for small recurrences, aiming to minimize radiation-related side effects in children [20,22,23]. However, there is a limited amount of data available on the use of proton therapy (PT), where published data mostly comprises case reports involving various tumor types and different age groups [24,25]. Treatment with protons, and particularly with pencil beam scanning (PBS-PT), allows a significant reduction in the integral dose in surrounding healthy tissues due to the distinctive physical properties of protons in contrast to conventional photon RT [26]. PT therefore has a clear benefit in the treatment of brain tumors, particularly among children. In fact, improved neuropsychological outcomes have been observed in pediatric patients with brain tumors after PT [27,28].

In recent decades, therapeutic advances have increased the life expectancy of pediatric cancer patients, emphasizing the crucial need to minimize treatment-related comorbidities, in turn improving patient quality of life.

The aim of this study was to report the clinical and patient-reported outcomes of children and adolescents with intracranial meningioma treated with PBS-PT at our institution.

## 2. Material and Methods

The institutional database was queried for meningioma patients treated with PBS-PT between 1996 and 2022 at the Paul Scherrer Institut. Out of a total of 207 patients, 10 (4.8%) were children or adolescents aged less than 18 years. All children had to have had either histologically confirmed meningioma (all World Health Organization (WHO) grade) or presumed meningiomas based on CT/MRI imaging and only underwent the delivery of proton therapy with no photon component.

The medical records of the identified children were retrospectively reviewed to collect demographic and clinical data (age, sex, referral country, presentation symptoms, tumor location, neurofibromatosis type 2 status, multiplicity, WHO histological grade, and Simpson [29] grading).

As part of a bigger study, this analysis received ethics approval (EKNZ 2022-00773). All patients were treated with PBS-PT with or without surgery. Children who could not remain calm or were unable to stay still during treatment were administered sedation and were closely monitored by a pediatric anesthesiologist throughout the procedure [30]. The total prescribed dose was based on the tumor grade in accordance with the WHO classification that was valid at the time of presentation [31], as well as other prognostic factors. Radiological criteria classification of the tumor [32] was applied to one patient who did not have surgery. A relative biological effectiveness factor for protons of 1.1 (relative to ^60^Co) was employed, and proton doses were expressed in terms of gray equivalent (Gy (RBE)) [33].

Treatment volumes were contoured by a radiation oncologist according to international guidelines [20,21,34,35]. Gross tumor volume (GTV) was defined as the visible lesion on contrast-enhanced T1-weighted images. Clinical target volume (CTV) was created for WHO grade 2 tumors by adding an additional margin to the GTV, and also considering the inclusion of initial tumor extension before surgery, hyperostotic bone changes, and dural thickening. Additional expansion of 4–5 mm was given to the CTV to define the planning target volume (PTV) depending on the immobilization technique (bite-block vs. thermoplastic mask).

Time to an event was calculated from the start of the treatment. As no gross total resections (GTR) were achieved, local failure (LF) was considered as the growth of residual tumor tissue before PT. In-field and marginal LF were defined as those developing inside or outside the 90% isodose, respectively, but still within the irradiation area. Out-of-field LF was defined as the those developing outside the irradiation area. Acute and late toxicity were defined as adverse events developing before and after 3 months from the start of the treatment. The evaluation of acute and late side effects was carried out according to the Common Terminology Criteria for Adverse Events (CTCAE) in its different versions over the years [36].

Since most of the children/adolescents were living far away, follow-up was performed in their local centers of origin by the referring medical team. Therefore, clinical follow-up reports, DICOM images, and radiological reports were periodically transferred to our system at the request of our Study and Research Office. All follow-up MRIs were reviewed and co-registered with the ones used for treatment planning or, alternatively, the last one available prior to radiotherapy treatment. To evaluate the best radiological response after treatment, Response Evaluation Criteria in Solid Tumors (RECIST) v1.1 criteria were applied [37]. The assessment of progression was obtained from the radiological reports and neuro-oncology committees of the referral centers.

Quality of life (QoL) before PBS-PT was assessed using PEDQOL surveys [38]. This validated instrument evaluates physical, emotional, social, and school functioning domains. The obtained score ranges from 0 to 100, with higher scores representing better QoL. The assessment of functionality after the end of treatment was carried out by in-house designed questionnaires. These were sent regularly on an annual basis and addressed to the children from the age of thirteen. For younger children, the questionnaires were addressed to parents/caregivers. Since the questionnaires changed slightly over the years, only those variables that remained constant in all the questionnaires were evaluated: need for educational support in case of attendance of regular education, need to attend a special school, social skills impairment with family or friends, and assistance with basic activities of daily living. All available questionnaires were reviewed for each child, and data were analyzed for the first year of follow-up, as well as two to three years after the end of treatment. A sample questionnaire can be found in the Appendix A.

Statistical analysis consisted of the Kaplan–Meier method to analyze local tumor control (LC) and overall survival (OS). All statistical tests were performed with Stata IC software (v.16), StataCorp LLC, College Station, TX, USA.

A literature search in PubMed database was performed using keyword combinations “meningioma”, “pediatric” or “children”, and “radiation therapy” or “proton therapy”. For the qualitative synthesis of the literature review, studies with cohorts comprising a minimum of 10 patients, published from the year 2000 onwards, and written in the English language were included. Some articles were excluded as they did not align with the description although they were found under the specified keywords. Reference lists of those articles were further examined for additional references.

## 3. Results

### 3.1. Patient Characteristics

The median age of children/adolescents at the time of receiving PT was 13.9 years (range, 3.2–17.2), and four (40%) of the ten children were in the first decade of life. There was a male predominance with a male to female ratio of 4:1. Seven meningioma (70%) were located on the skull base, two (20%) on the optic nerve sheath, and one (10%) on the convexity. Three children (30%) had evidence of associated type 2 neurofibromatosis (NF2), and all of them had multiple meningiomas.

The clinical manifestations observed at diagnosis were in all cases multiple and heterogeneous. Increased intracranial pressure occurred in four children (40%), exophthalmos in three (30%, optic nerve sheath n = 2, left cavernous sinus = 1), sensorial deficit (visual or hearing) in four (40%), and decline in school performance in two patients (20%).

### 3.2. Treatment Characteristics

No gross total resections were achieved. Most of the resections performed were classified as Simpson grade IV (n = 8, 80%). Only one (10%) child was biopsied (Simpson V), and another (10%) child did not undergo any surgery. Prior to PT, the median number of surgeries performed was two (range, 1–3). Out of the nine children who underwent surgery, over half (n = 5, 55.6%) required more than one intervention.

Baseline neurological symptoms before PT were observed in nine patients (90%), of which 66.7% were tumor related, 11.1% surgical related, and 22.2% both tumor and surgical related. Detailed information about the neurological status before PT and the location of the irradiated tumor is detailed in Table 1.

The histopathological analysis showed that out of the nine operated meningiomas, two (22.2%) were classified as WHO grade 1 tumor and seven (77.8%) as WHO grade 2. No WHO grade 3 meningiomas were observed. In terms of the histological subtype, three (33.3%) were identified as clear cell meningioma, two (22.2%) as atypical meningioma, one (11.1%) as meningothelial, one (11.1%) as fibrous, one (11.1%) as mixed with papillary component, and one (11.1%) merely as meningioma.

Six children (60%) were treated at initial diagnosis (four in a postoperative adjuvant setting and two with exclusive PBS-PT), while four (40%) were treated at recurrence or disease progression. The median time between diagnosis and PT was 6.3 months (range, 2–30).

The median volumes of the GTV and PTV were 31.3 cc (range, 5.8–241.7) and 76.1 cc (range, 27.5–528.2), respectively. The median prescribed total dose was 59.4 GyRBE (range, 50.4–64.0 GyRBE). The median number of fractions given was 32 (range, 28–34).

### 3.3. Clinical and Patient-Reported Outcomes

With a median follow-up time of 71.1 months (range, 2.5–249.7), the estimated 3- and 5-year LC were 58.3% (95%CI 18.0–84.4) and 19.4% (95%CI 9.3–56.3), respectively. Five children (50%) experienced a LF within a median time of 32.4 months (range, 17.7–55.4). Of those, four (80%) were in-field and one (10%) marginal. No out-of-field or distant failures were observed.

Two children required multiple salvage treatments after tumor progression. Three children underwent re-resection, three received systemic treatment, and re-irradiation was performed in two. Detailed information regarding these therapies can be found in Table 2.

During follow-up, a reduction in over 30% in the sum of the longest diameters of the tumor was observed in two (20%) patients, which was assessed as a partial response. The median diameter decrease was 7.8 mm (range, −4.14–25.18). However, most patients (n = 8) had stable disease as a best radiological response after irradiation before tumor progression (n = 5).

The estimated 3- and 5-year OS was 100%. One patient died of T-cell lymphoma 127.1 months after PBS-PT.

Common acute toxicity events consisted of alopecia (60%), radiation dermatitis (50%), conjunctivitis (20%), and fatigue (20%). No grade 3 or more acute toxicity was observed. The most common late toxicity events were hypopituitarism (50%) and alopecia (20%). Four (40%) children experienced late grade 2 events, which are detailed in Table 2. Except for one case of cataract that required surgery, no grade 3 or higher late toxicity events were reported.

A total of five parents and four patients completed the E1 (baseline) PEDQOL survey. Before undergoing PT, patients rated their QoL lower than their parents across various domains, including autonomy, body image, cognition, family social functioning, and subjective well-being. The patients’ QoL scores with regard to emotional functioning, physical functioning, and peers social functioning were higher compared to their parents scores (Figure 1).

With regard to the functionality assessment after the end of treatment, the median number of questionnaires available per patient was two (range, 1–6), while median time until the last available questionnaire was 37.6 months (range, 5.75–112.6). During the follow-up, four children (44.4%) pointed to the need of educational support, three (33.3%) experienced social impairment, two (22.2%) attended a special school, and three (33.3%) required treatment for DBA (Table 3). The responses collected from the filled questionnaires, during the first year and 2 and 3 years after the therapy, are represented on Figure 2.

## 4. Discussion

Hereby, we reported the clinical and patient-reported outcomes following PBS-PT of a small cohort of children and adolescents with intracranial grade WHO 1-2 meningiomas. We observed a modest 5-year LC rate of 19.2%, which is worse than that observed in adults and highlights the unique characteristics of this entity. Our 5-year OS of 100% is, however, encouraging and might be reflective of the improvement of therapies and effectiveness of salvage treatment over the years. Of note, all children and adolescents presenting with treatment failures were aggressively salvaged (Table 2). The excellent tolerance of the treatment in terms of acute toxicity and absence of grade 3 long-term toxicity supports the safety of the treatment. The need for additional support at school and DBA in some children underscore the vulnerability of this population.

Our observed prevalence of 4.8% over a period of 23 years is in line with the published literature. Meningiomas are rare tumors in children and adolescents, accounting approximately for 0.4–4.6% of all brain tumors in these age groups [16,39,40].

The majority (80%) of our patients were male (Table 2). The published gender ratio is quite inconsistent. In various case series, a male predominance has been usually described for young meningioma patients [18,41,42,43]. Rochat et al. found a male preponderance in the younger age group with a gradual shift towards a female predominance as the age group increased [44]. Similarly, a retrospective analysis using the North American Surveillance, Epidemiology and End Results (SEER) database found that the group of children and adolescents had an equal male–female ratio and that only pre-pubertal children (0–11 years in boys and 0–8 years in girls) showed a male predominance [39].

In our series, all but one assessable patient presented with a non-benign tumor (Table 2). It has been described that the proportion of high-grade meningiomas is higher in children and adolescents in comparison to adults [8,21,45]. The lower proportion of grade 1 meningiomas observed in our small cohort is thus in line with the published data (Table 4).

Maximal resection remains the standard of care and there is a lack of high-quality data regarding the role of adjuvant RT (Table 4). Due to the increased vulnerability to potential adverse effects in children, RT has been typically reserved for malignant meningiomas or tumor recurrences, particularly when complete surgical removal is not achievable [17,22,46]. Therefore, one would expect a lower proportion of patients treated with RT among children and adolescents. However, it is worth mentioning that the SEER study found radiation therapy as part of the therapy in 14.6% of the children/adolescent group (0–21 years), higher proportion compared to the young adults (22–45 years, 12%) and older adults (>45 years, 8.5%) groups [39] that may reflect the more aggressive nature of these tumors when compared to their adult counterparts.

In the present study, we observed five local failures (50%) after a median time of 32.4 months, with an estimated 5-year recurrence-free survival of less than 20%. The pronounced difference in local control rates could be explained by the presence of a higher proportion of unfavorable factors in this highly selected cohort compared to other series (Table 4). The Paul Scherrer Institut had been the first facility delivering PBS-PT, presumably concentrating highly selected patients with complex tumors. It should be noted that no resections lower than Simpson IV were achieved for our patients, in the context of challenging locations and voluminous tumors, 70% were WHO grade 2, 40% were recurrent tumors after upfront therapy, and 30% presented multiple tumors in the context of neurofibromatosis type 2. The extent of surgical resection, WHO grade 3, and neurofibromatosis type 2 have been associated with worse prognosis [6,9,46,47,48]. The meta-analysis conducted by Kotecha et al., which included a total of 677 children and adolescents with meningioma who underwent surgery, reported a significantly lower 5-year relapse-free survival (RFS) in patients with subtotal resection (STR, 46.0% vs. 85.8% after gross total resection, GTR) and WHO grade 3 tumors (40.7% vs. 81.2% in WHO grade 1) [8]. The study conducted by Wang et al., which only included higher grade meningiomas, of which 65% had Simpson IV resection, observed tumor recurrence in 50% of the patients, with a mean recurrence time of 22.4 months after the surgery. In their series, 39% of the patients received RT, and 44.4% of them recurred. However, details regarding time to recurrence were not available for all cases [49].

Despite the unfavorable local control rates found, we observed an encouraging estimated 5-year OS of 100%, which is higher compared to previous data available in the literature. Kotecha et al. estimated a 5-year OS of 90.3% in their meta-analysis. The more recent analysis from Dudley et al. [39] describes a 4.5% mortality in the youngest age subgroup (0–21 years). Tumor recurrence followed by perioperative events have been described as a relatively common cause of mortality in these patients [10,11,16,41]. Our findings may be related to an improvement in the surgical techniques over recent years but particularly to the effectiveness of the salvage therapies, which included re-resection, re-irradiation, and systemic treatment. Together, both assumptions highlight the critical value of the management of these patients by multidisciplinary teams in experienced referral centers. Although systemic therapies have been the most employed salvage therapy for our patients (Table 2), the evidence of these strategies remains limited [50,51,52]. Furthermore, the absence of malignant meningiomas in our cohort probably contributed to this excellent OS rate.

Radiosurgical treatment has been preferred in the site of the unresectable recurrence to avoid radiation adverse effects in the young population [23,46,53]. The study conducted by Mishra et al. evaluated the use of Gamma Knife radiosurgery (GKRS) for CNS tumors in 40 children. Out of a total of 61 tumors, 20 were meningiomas, making it one of the largest series of pediatric meningiomas treated with radiation therapy documented thus far. Despite the relatively short mean follow up of 15 months, all cases showed either reduction or stabilization of the tumor volumes [54] 

The favorable local control rates of PT in the adult population have been described in a recent review [55], and this technique is included in the most recent National Comprehensive Cancer Network guidelines for meningioma [56]. In those situations where the characteristics of the target volume do not allow stereotactic treatment, protons would offer a potential benefit compared to photon treatment in the pediatric and adolescent population due to the reduction in the integral dose to the surrounding tissues, which may associate a reduction in long-term side effects such as neurocognitive detriment and secondary tumors [28,57,58].

In the present series, treatment with PT was well tolerated with no acute or late grade 3 or higher toxicities observed, except for one case of cataract that required surgical intervention. The most prevalent long-term adverse effect was pituitary dysfunction with the need for replacement therapy, observed in 40% of patients, which may be related to the location of 70% of the tumors in the skull base. These findings are in line with the incidence described in previous studies of proton therapy administered in children or young adults with brain tumors [59,60].

Before proton therapy, patients reported poorer QoL than parents in most domains. In contrast, it has been described both that parents tend to over-report their children’s QoL problems and that children sometimes underestimate their problems due to neuro-psychological denial. [61,62]. Noteworthy, given the median age of these patients, it is reasonable to expect them to have enough insight to perceive their quality of life in domains such as autonomy, body image, social functioning, and subjective wellbeing.

Between 22% and 44% of our children and adolescents experienced a noticeable impact on their functionality at some point during follow-up. The highest proportion of affirmative responses for the analyzed variables were registered two years after PT. After three years, only one patient reported the need for assistance with daily basic activities, although another three patients did recur at this time-point. Similarly, in a prospective study conducted at our institution by Kroeze et al. [63], which focused on adult brain tumor patients, a temporary decrease in global QoL was observed after treatment, followed by an improvement starting from one year onwards. Higher quality scores have been seen to significantly correlate with a longer time from treatment [64]. The study conducted by Kuhlthau et al. reported the health-related quality of life (HRQoL) outcomes in a PT-treated pediatric brain tumor cohort showing a progressive improvement in the score from baseline to 3 years after PT [65]. It is important to note that the number of completed questionnaires in our cohort decreased over time. Moreover, other factors such as abnormal neurological status prior to proton therapy (caused by the primary tumor or prior surgical procedures) and tumor progression may have had an influence on the studied variables, which significantly restricts the interpretation of the observed outcomes.

There were several limitations of our study. Firstly, the study design was retrospective in nature and thus lacked complete data for certain variables such as full imaging follow-up datasets. The small sample size of 10 children/adolescents limits the interpretation of these data, and caution should be exercised so as not to over-interpret them. In addition, methylation family data could not be obtained for the meningiomas of these patients.

Despite the aforementioned limitations, this is, to our knowledge, the largest series to report the long-term results of PBS-PT as part of the treatment of pediatric and adolescent patients with intracranial meningioma.

**Table 4 cancers-15-04447-t004:** Publications on pediatric/adolescent meningioma with cohorts of 10 patients or more from 2000 onwards.

Author Year Reference n.	n	FU (m)	Mean Age (y) (Range) M:F Ratio	Location/NF Status/ Resection (%)/WHO Grade	Radiation Therapy N (%)/Indication/Dose, Technique	Outcomes ^R^ Recurrence (%)/† Mortality (%)/ ° RT Outcome
Amirjamshidi 2000 [66]	24	130.2	9.47 (2–17) 11:13	20 IC, 1 IO 5 NF (excluded) 21 GTR (87.5%) -	1 (4.2%) Irresectable recurrence -	R: 6 (25%) † 1 (16.6%) -
Lund-Johansen 2001 [67]	27	-	14.8 (0–20) 16:11	20 IC 5 NF 19 GTR (70.4%) 27 G1 (100%)	3 (11.1%) Residual tumor, recurrence 2 GK, cRT	R: 8 (29.6%), 2 after GTR † 3 (11 %), 1 surgery-related -
Im 2001 [46]	11	108	8 (0.5–14) 5:6	10 IC, 1 IO 1 NF 8 GTR (72.7%) -	4 (36%) 2 residual and 2 recurrences GK, cRT	R: 3 (27.2%) † 1 (9%) °3 SD, 1 death not tumor-related.
Zwerdling 2002 [68]	18		11 (1.6–17) 8:10	13 IC, 4 IO - 11 GTR (61%) 4 G3 (22%)	4 (22.2%) 2 postoperative, 2 definitive -	R: 3 (16.6%) † 2 (11%) °2/4 died, after 4 and 16 m survival
Rochat 2004 [44]	22	192	5 (M),11.5 (F) (0–14) 8:14	All IC 3 NF 15 GTR (68.2%) 20 G1 (90%)	8 (36%) - -	R: 9 (40.9%) † 13/22 tumor-related (59%). OS rate 38%. -
Rushing 2005 [69]	87	68.5	14 (0.42–20) 52:35	81 IC 9 NF2 53 GTR (62%) 62 G1 (71.3%), 21 G2, 4 G3 (4.6%)	4 postoperative (4.6%) Residual tumor, recurrence -	R: 12 † 7/62 (11.3 %) ° 2/4 died
Tufan 2005 [47]	11	72	12.7 (1.2–17) 6:5	All IC 1 NF 8 GTR (73%) 6 G1 (54.5%), 2 G2, 3 G3 (4.6%)	1 postoperative (9.1%) Recurrence -	R: 3 (27.2 %) † 3, (2 perioperative) -
Caroli 2006 [42]	27	108	11.3 (0.5–16) 2.8:1	- - 21 GTR (77.8%) 3 G3 (11.1%)	-	R: 13% † 1, preoperative -
Greene 2008 [70]	20	20	13 (3–20) 11:9	15 IC, 2 IO 5 NF2 - 2 G3 (10%)	4 (20%) - Median 59.4Gy (range 52.2–59.4)	R: 4 (20 %) † 3 (15 %), 1 tumor-related -
Arivazhagan 2008 [45]	33	23.4	14.7 (5–18) 19:14	32 IC, 1 IO 3NF 22 GTR or near (66.7%) 29 G1 (87.9%), 2 G2, 2 G3 (6%)	4 adjuvant (12%) Atypical incomplete resection and anaplastic -	R: 6 recurrences/regrowth (18.1%) † 3 (9%) -
Liu 2008 [7]	12	-	9.9 (0.5–15) 1.4:1	12 IC 0 NF - 0 G3	-	- † 3.3% (Including patients from 8 series reported in the literature) -
Harold Lee 2008 [19]	14	150	13.8 (6–18) 8:6	All ONS 4NF2 4 GTR (28.6%), 7 STR, 2 Bx, 1 none -	3 (25%) 2 only biopsied, 1 after re-resection of recurrence Conformal RT	R: 7 (50%) alive with disease, 7 (50%) alive without disease † 0 ° 1 alive without disease (after 31y FU), 2 alive with disease (after 3m FU)
Gao 2009 [16]	54	62.7	13.1 (2.8–18) 29:25	52 IC, 1 IO 5 NF2 39 GTR (72.2%) 18.5% G2–3	7 adjuvant (13%) Incomplete resection, anaplastic -	R: 10 (30.3 %) † 9 (16.6 %), 2 perioperative (3.7%) ° 3 recurrences (42.8%)
Li 2009 [71]	34	48	Med 10 (2–17) 29:30	34 IC - 20 GTR, 11 STR 6 (17.6%) G2–3	15 (44.1%) Residual, malignant, and recurrent tumors	R: 7 (20.5 %) † 6 (17.6 %) ° 3 recurrences (20%)
Menon 2009 [72]	38	56.9	15.53 (2.5–<20) 20:18	31 IC, 2 IO 11 NF (28.9%), 2NF2 20 GTR (48.8%) 30 (73.2%) G1, 9 G2, 2 G3 (4.9%)	- Adjuvant therapy in G 2 and 3 variants -	R: 7 (18.4%). † 1 (2.6%). -
Mehta 2009 [41]	18	73.2	12.81 (0.75–18) 1.57:1	18 IC 2 NF2 18 GTR (100%) 1 G3 (5.6%)	4 postoperative (22.2%) Aggressive pathology -	R: 2 after 2 and 5 years (14.2%). † 1 perioperative (0.1%) -
Lakhdar 2010 [22]	21	33	10.3 (2–16) 13:8	21 IC 1 NF 13 GTR (61.9%), 8 STR 6 G3 (28.6%)	5 (23.8%) postoperative 4 Residual tumor, 1 after recurrence 50–60 Gy	R: 7 (33.3%) † 2 (9.5%) -
Thuijs 2012 [9]	72	57.6	13 (0–18) 39:33	51 IC, 4 IO 13 NF2 (18%) 35 GTR (48.6%), 29 STR 53 G1 (73.6%), 13 G2, 6 G3 (8.3%)	15 (20.8%) postoperative Recurrence, atypical, or malignant Mean dose 42.75 Gy (13–60)	R: 26 (36%) 12/46 (GTR) recurrences, 14/43 (STR) re-growths † 16.1% 1y OS 96% (n = 50) 5 y OS 83.9% (n = 31) 1y PFS 84.3% (n = 51) 5 y PFS 55.6% (n = 36)
Santos 2012 [73]	15	60	12 (4–18) 9:6	14 IC 3 NF2 (20%) 14 GTR (93.3%) 11 G1 (73.3%), 4 G2	1 (6.6%) Recurrence SRS	R: 5 recurrences (33.3%), 4 re-operated - -
Wang 2012 [49]	23	70	12.1 (2–18) 18:5	20 IC, 3 SP 3 NF2 (13%) 11 GTR, 11 STR, 1 missing 15 G2 (65%), 8 G3 (35%)	9 (39.1%) - -	R: 10/20 (50%) in a mean time of 22.4m † 1 postoperatively, 2 lost FU 10y-PFS rate 54.5% 10 y-OS rate 63.3% ° 4/9 recurred (44.4%)
Ravindranath 2013 [43]	31	46.2	14.15 (0.7–<18) 22:9 1:2.4	All IC 2 NF2 (6%) 26 (83%) GTR, STR 5 20 G1 (64.5%), 11 G2–3	9 (29%) Higher grade and recurrence 45–50 Gy	R: 20 recurrences (64%), 15 after GTR. † 1 death (3%), 3 lost -
Grossbach 2017 [12]	39	-	14.1±4.57 (1–20) 15:24	28 IC, 3 IO, 8 SP 8 NF (20.5%), 4 NF2 27 G1 (69.2%), 10 G2, 2 G3 (5.1%)	13 (33.3%) Recurrences, aggressive pathology, multiplicity, STR and inoperability. Fractionated RT, 52.7–64.8Gy. 1 SRS.	R: 15/36 who underwent resection (42%) - -
Huntoon 2017 [13]	15	-	12.8 (1–18) 6:9	14 IC, 1 SP - (excluded) - 8 G1 (53.3%), 5G2, 2 G3 (13.3%)	2 (13.3%) G3 1 cRT, 1GK	- - ° 1 alive, 1 died.
Liu 2017 [23]	19	40.9	14.7 (7–18) 9:10	Infra-tentorial 2 NF2 GTR 14 (73.7%), 5 STR G2–3 (26.3%)	8 (42%) 4 GK, 2 as primary treatment, 2 for recurrent/STR 4 cRT G2,3	R: 5 (26.3%) - ° 2 cRT had quick recurrence
Wu 2017 [17]	14	66.1	11.1 (4–16) 9:5	14 SP - (excluded) 11 GTR (78.6%), 2 STR, 1 PR 5 G2	3 (21.4%) Recurrences, following re-resection Dose not specified	R: 4 (28.6%). 1 after GTR, 3 in STR and PR. 4 in G1, 5 in G3. - °2 recurrences after re-resection + RT
Fan 2017 [74]	32	48	13.7 (2–18) 17:15 1.13:1	All IC - 19 GTR (48.7), 13 STR 16 G1 (50%), 5 G2, 11 G3 (34.4%)	1 (3%) Recurrence Focal salvage RT	R: 9 (28.1%), 4 after GTR, 5 after STR † 1, 9 m after the diagnosis of recurrence -
Dudley 2018 [39]	381 *7464* *51,303*	35	0–21, *22–45,* *>45*	328 IC (86.1%), 50 SP - 163 GTR (43.4%), 24 STR, 67 Bx, 122 any	55 (14.6%) *883 (12.0%)* *4313 (8.5%)*	† 4.5% all-cause mortality *4.5%* *24.6%*
El Beltagy 2019 [11]	39	38.5	8.19 21:18	36 IC, 3 SP 4 NF2 (10.2%) 28 GTR (71.8%), 8 STR, 3 Bx 16 G1 (41%), 11 G2, 12 G3 (30.8%)	19 (48.7%) Higher grade (14 after GTR, 5 STR) -	R: 5-year EFS (event FS) 85.6% † 5 (12.8%, tumor-related) 5-year OS 87.8% ° 1 R 4m after GTR+RT
He 2020 [6]	39	54.4	Med 13 (1–18) 1.3:1	36 IC, 1 ONS, 2 IO 3NF, 1NF2 28 GTR (71.8%), 11 STR 26 G1 (66.7%), 10 G2, 3 G3 (7.9%)	8 (20.5%) Residual tumor, high grade, recurrence GK	R: rate 41.9% mean time to R 20.2m (median 12m) 5 years EFS 55.1% -
Isikai 2020 [48]	23	123.6	13.1 (-sd ±4.4) 12:11	19 IC, 4 SP 6 NF (26.1%), 5 NF2, 1 NF1 19 GTR (70.4%), 8 STR 15 G1 (56%), 9 G2, 3 G3 (13%)	5 (21.7%) STR after recurrence	R: 10 (43.5%), 4 after GTR, 6 after STR † 3 (13%) 1y OS 91% 10y OS 86% -
Jain 2020 [75]	37	24	14 (2.5–20) 20:17	31 IC, 6 SP 16% NF >50% GTR >50% G1	4 (10.8%) Residual tumor, high grade, recurrence	R: 8 (33.3%; re-surgery) - -
Thevandiran 2020 [14]	10	60	10.5 (1–18) 4:6	9 IC, 1 SP 1 NF2, 1 NF1 7 GTR (70%), 3 STR 6 G1 (60%); 4 G2.	No RT	R: 1 (11.1%), after 12m - -
Liu 2021 [18]	40	82.1	10.8 (0–15) 1:1.11	38 (95.2 %) IC, 1 ONS, 1 SP (2.4%) 0 NF (excluded) 35 GTR (83.3%), 7 STR (17.5%) 28 G1 (66.7%), 9 G2, 5 G3 (11.9%)	2 (5%) G3 -	R: 9 (22.5%) † 5 (12.5%) -
Mishra 2022 [54]	20/61 neoplasms in 40 children	15	6–18	- 14 NF2 - -	GK (12.55 Gy mean marginal dose) 6 primaries, 14 secondaries (after surgery/cRT) -	° 6 reduced sizes, 14 SD -
Opoku 2022 [10]	10	24	9.3 (4–16) 6:4	9 IC, 1 foramen magnum 0 NF (excluded) 10 (100%) GTR 3 G1 (12.5%), 4 G2, 3 G3 (12.5%)	No RT	R: 0 † 1 postoperative

M: male; F: female; FU: follow-up; sd: standard deviation;y: year; Med: median; IC: Intracranial; IO: (intra)orbital; ONS: optic nerve sheath; SP: spinal; NF: neurofibromatosis; G: grade (WHO); GTR: gross total resection; STR: subtotal resection; PR: partial resection; Bx: biopsy; cRT: conventional radiation therapy; GKRS: Gamma Knife radiosurgery; R: recurrence; † mortality; ° RT outcome; SD: stable disease; OS: overall survival; PFS: progression-free survival; EFS: event-free survival

## 5. Conclusions

Considering the favorable treatment tolerance and the absence of severe side effects, the present study supports the use of PBS-PT as a treatment option for intracranial meningiomas in children and adolescents. The observed outcomes highlight the aggressive nature of these tumors in younger age groups, with a higher likelihood of recurrence compared to their adult counterparts. Salvage therapy must be taken into consideration since it might contribute to a better survival after tumor progression as observed in our cohort. Further research is necessary to elucidate the role of radiation therapy in the pediatric population, particularly in cases involving incomplete tumor removal, high-grade tumors, or unresectable recurrences. Due to their low frequency, collaborative efforts among multiple institutions for data collection should be encouraged. Furthermore, by the increasing understanding of the distinct molecular characteristics of pediatric meningiomas, it is to be expected that advancements in targeted therapies will expand the available therapeutic landscape for this challenging disease.

## Figures and Tables

**Figure 1 cancers-15-04447-f001:**
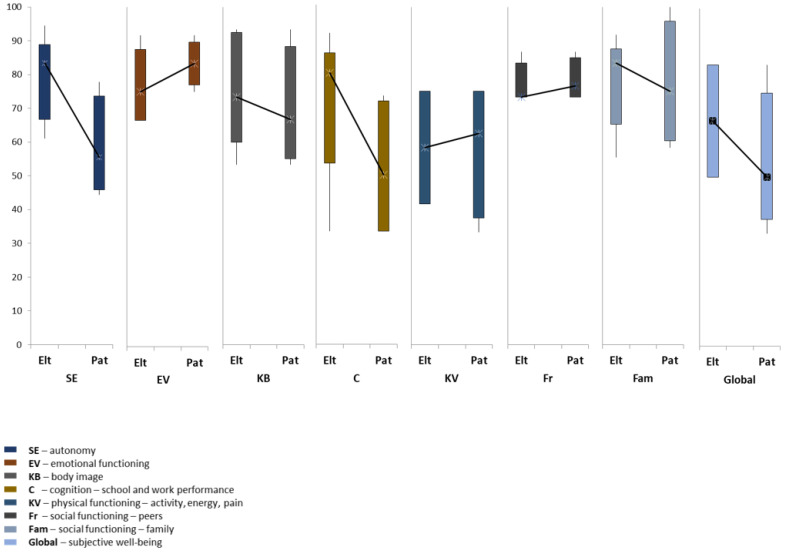
Parents (Elt) and patients (Pat) reported quality of life scores before proton therapy (PEDQOL, E1).

**Figure 2 cancers-15-04447-f002:**
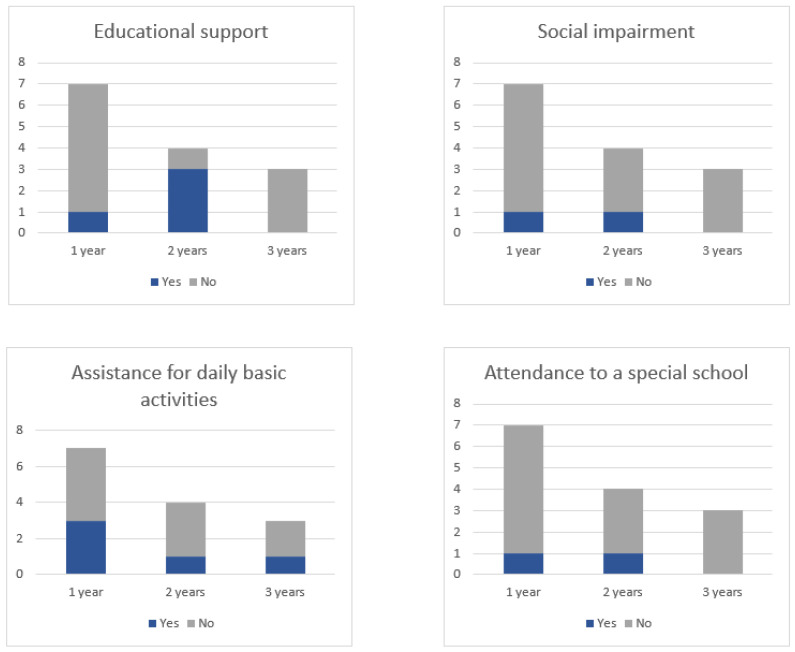
Patient reported outcomes based on in-house questionnaires for evaluating functional status during the 1st year and 2 and 3 years after PBS-PT.

**Table 1 cancers-15-04447-t001:** Detailed neurological status before PBS-PT.

Case ID	Symptoms	Main Cause	Irradiated Tumor Site
1	Behavior changes	Tumor-related	Right clinoid process
2	Decline in school performance Left-sided exophthalmos Left-sided severe hearing loss * Right-sided distal radial paresis * Right-sided visual impairment ~	Tumor-related	Left retro-bulbar/pterygoid
3	Left-sided exophthalmos and ptosis Left-sided trigeminal palsy with secondary left corneal dystrophy	Both tumor- and surgery-related	Left cavernous sinus, cerebellopontine angle and left temporal region
4	Seizures Left-sided hemiparesis with the need of orthopedic cast for walking and wheelchair for long distances Right-sided VI and VII cranial nerves palsy Communication skills impaired	Surgery-related	Right posterior fossa and cerebellopontine angle
5	Left-sided limited gaze elevation in adduction Left-sided temporal visual field defect Right-sided amaurosis	Tumor-related	Suprasellar
6	Right-sided VII palsy Right-sided postoperative deafness Right-sided hemiplegia with the need for a lower limb cast for long distances *	Both tumor- and surgery-related	Right cerebellopontine angle
7	Left-sided visual impairment Left-sided exophthalmos and ptosis Diplopia Left-sided mild neurosensory hearing loss *	Tumor-related	Left optic nerve sheath
8	Hearing impairment Decreased sensitivity in the right territory of V3	Tumor-related	Right cavum Meckel, petrous bone, prepontine cistern
9	Left-sided hearing loss	Tumor-related	Left carotid space, jugular foramen, middle ear, and auditory canal
10	None	-	Right parietal

* Secondary to a different tumor. ~ In the context of microphthalmos with persistent hyperplastic primary vitreous.

**Table 2 cancers-15-04447-t002:** Patient characteristics, treatment details, and outcomes.

ID	Sex	Age (Y)	Location	NF2	WHO Grade	Simpson Resection	T Intent	Dose (GyRBE) fr (n)	GTV (cc)	PTV (cc)	FU Time (m)	Acute Tox, Grade	Late Tox, Grade	Time to LF (m)	LF (Type)	Salvage Therapy	Status
1	M	7	Skull base	No	2	IV	Postop.	64.0 32	32.2	81.4	249.7	Alopecia, 1 Dermatitis, 1	Hypopituitarism, 1 Endocrinol other, 1	55.4	Yes (in-field)	Hydroxyurea (Litalir)	Alive
2	M	13	Optic nerve sheath	Yes	2	IV	Postop.	54.0 30	46.2	101.0	175.3	Dermatitis, 2 Conjunctivitis, 1	Cataract, 3 Hypopituitarism, 2	51.8	Yes (marginal)	Four debulking surgeries 45 Gy in 25 fractions of 1.8 Gy Bevacizumab (Avastin)	Alive
3	F	3	Skull base	No	1	IV	Postop.	59.4 33	9.7	36.7	132.7	Alopecia, 1 Appetite loss, 1	Hypopituitarism, 2	32.4	Yes (in-field)	Surgery	Alive
4	M	6	Skull base	No	2	IV	Postop.	57.6 32	24.5	58.2	127.1	Alopecia, 1 Dermatitis, 1	Hearing loss, 1 Hypopituitarism, 2		No		Death *
5	M	4	Skull base	No	2	IV	Postop.	61.2 34	30.5	70.9	104.4	No	Hypopituitarism, 2 Vascular, 2	28.4	Yes (in-field)	Avastin/Irinotecan Sandostatin Sunitinib Temodal Embolization + surgery TTF/Optune	Alive
6	M	14	Skull base	Yes	2	IV	Savage	59.4 33	34.4	170.2	37.8	Alopecia, 1 Fatigue, 1 Nausea, 2	No		No		Alive
7	M	17	Optic nerve sheath	Yes	NA	NA	Definitive	50.4 28	9.6	27.5	26.0	Conjunctivitis, 1	Alopecia, 1		No		Alive
8	F	15	Skull base	No	2	IV	Savage	60.0 30	5.8	64.7	17.7	Alopecia, 1	Alopecia, 1	17.7	Yes (in-field)	GK, 16 Gy to 53% isodose	Alive
9	M	16	Skull base	No	1	V	Definitive	50.4 28	241.7	528.2	5.8	Dermatitis, 2 Oral mucositis, 2	No		No		Alive
10	M	14	Convexity	No	2	IV	Postop.	59.4 33	142.3	454.0	2.5	Alopecia, 2 Dermatitis, 1 Fatigue, 1			No		Alive

Abbreviations: ID: identification; Y: years; NF2: neurofibromatosis type 2 positive; PT: proton therapy; GTV: gross tumor target volume; PTV: planning target volume; Tox: toxicity; FU: follow-up; LF: local failure; NA: not applicable; RT: radiation therapy; GK: Gamma Knife; m: months; n: number; cc: cubic centimeter; TTF: tumor treating fields. * This child died due to T-cell lymphoma.

**Table 3 cancers-15-04447-t003:** Questionnaire’s variables with positive answer at some point during the follow-up.

ID	Educational Support	Special School	Social Impairment	Assistance for DBA
1	No	No	Yes	No
2	Yes	Yes	No	No
3	Yes	No	No	No
4	Yes	No	Yes	Yes
5	Yes	Yes	Yes	Yes
6	No	No	No	No
7	No	No	No	No
8	No	No	No	No
9	No	No	No	Yes
10	NA	NA	NA	NA

Abbreviations: DBA: daily basic activities; NA: not available.

## Data Availability

The data presented in this study are available upon request from the corresponding author.

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
