# Peer review of "Quality of Life, Clinical, and Patient-Reported Outcomes after Pencil Beam Scanning Proton Therapy Delivered for Intracranial Grade WHO 1–2 Meningioma in Children and Adolescents"

_cancers, 2023, doi:10.3390/cancers15184447_

Round 1

Reviewer 1 Report

The authors developed a very interesting topic about the clinical and patient-reported results of children and adolescents

diagnosed with intracranial meningioma who received pencil beam scanning proton therapy (PBS-PT).

The study included a total of 10 patients under the age of 18, out of a cohort of 207 intracranial meningioma patients

treated with PBS-PT. The researchers assessed acute and late toxicities using  CTCAE and evaluated the quality of life (QoL)

using PEDQOL questionnaires before PBS-PT.

Educational, functional, and social aspects after treatment were carefully assessed.

Results showed that 50% of the children developed local failure at a median time of 32.4 months after PBS-PT.

Overall the authors demonstrated that PBS-PT appears to be a viable treatment option for children with intracranial meningioma, as it provides outcomes comparable to other radiation therapy approaches. Though patients reported lower QoL compared to their parents before treatment, functional status profiles were generally acceptable, and severe adverse radiation-induced events were rare.

Despite the small sample size of 10 children/adolescents, due to the rarity of disease, this investigation is well designed and offers a valuable contribution to current literature.

I suggest to describe how the systematic review of the literature was performed, including the Flow chart of Search strategy (PRISMA).

Author Response

We would like to thank the reviewer for the positive comments and overall manuscript-assessment.

Unfortunately, since the main aim of the study was not to perform a systematic literature review, PRISMA guidelines were not applied. The table with the literature synthesis was made to facilitate the contextualization of the topic and the comparison of our results with those available in the literature. We however fully agree with the reviewer’s assessment on the need to include information on how the literature review was performed and how the Table was populated. As such, we have modified the MS and provided the reader with such information.

Reviewer 2 Report

This is a rather small retrospective case series of children and adolescents with WHO grade 1-2 meningiomas who underwent pencil beam scanning proton therapy. Quality of life, Clinical, and patient-reported outcomes were presented. due to the rarity of meningiomas in this age group, only few previous reports exist concerning RT in young patients with intracranial meningiomas. One case in this case series did not have a histologically verified meningioma. This case can be excluded from the analysis. However, it is the first case series of PBS-PT in young menigioma patients. Therefore I would basically recommend this paper for publication after revision. 

minor editing required

Author Response

We would like to thank the reviewer for his/her positive comments and for his/her astute comments.

Point 1

Indeed, two of the patients had an optic nerve sheath meningioma (ONSM) and we did not have histologic confirmation for one of them since neither surgical resection nor biopsy of the lesion treated with proton therapy was performed, as it is common in the management of ONSMs. This was an adolescent with diagnosis of multiple meningiomas in the context of neurofibromatosis type 2.

Optic nerve sheath meningiomas are very rare but can be found in children and tend to be more aggressive than in adult population, with a prevalence of between 1:95,000 and 1:525,000 [Levin LA, Jakobiec FA. Optic nerve tumors of childhood: a decision-analytical approach to their diagnosis. Int Ophthalmol Clin. 1992].

Their diagnosis is based primarily on clinical and radiologic findings [Douglas VP et al. Optic nerve sheath meningioma. Curr Opin Ophthalmol. 2020] and MRI has become over the years the gold standard for diagnosis and has allowed to obviate in some cases the need for biopsy due to the high risk of complications [Bhupendra C. et al, Optic Nerve Sheath Meningioma, StatPearls, Internet, consulted on 26.08.2023, last update 8.08.2023].

In fact, other similar papers, such as that of Harlod Lee and colleagues [Harold Lee HB, Garrity JA, Cameron JD, Strianese D, Bonavolontà G, Patrinely JR. Primary optic nerve sheath meningioma in children. Surv Ophthalmol. 2008] who reviewed a cohort of children with ONSM, also included one case in which no resection or biopsy had been performed.

Noteworthy, meningioma papers usually contains histologically proven and radiologically diagnosed tumors. Given the rarity of this entity, and the potential of proton therapy as a therapeutic tool in the treatment of these patients, we would be in favour to keep this adolescent’s case included in the present analysis.

Point 2

Regarding the quality of the English language, a colleague from our center, a native English speaker, has kindly agreed to carefully review the redaction. Hopefully the modifications meet the criteria to be considered for publication in your journal.

Reviewer 3 Report

Meningiomas are characteristically slow growing, with extended survival compared to other malignant brain tumors, and typically present in the second through fourth decades of life in otherwise healthy individuals. Although less aggressive than their high-grade counterparts, these tumors are associated with neurologic and systemic symptoms and a diminished quality of life (QOL). These negative effects are attributed to both natural disease progression as well as treatment, which typically consists of maximal safe surgical resection followed at some point by radiation.

As in a recently published paper by the same research group the authors focused on “Long term outcome and quality of life of intracranial meningioma patients treated with pencil beam scanning proton therapy” Cancer 2023,15,3099 , the authors focused in this manuscript on the quality of life for intracranial grade WHO 1-2 meningioma in children. I was truly impressed by the very long follow-up time (median follow-up time 71.1 months) of the population of children suffering from meningiomas in this retrospective trial.

Overall, the paper is well written and easy to follow. Especially the discussion part is very detailed and the most relevant literature was cited. This makes the paper very attractive for the readers of cancers.

Author Response

We would like to thank the reviewer for her/his positive comments and are happy that this work contributes to the available literature on this rare entity in the pediatric and adolescent population.